

# Deep divergence of Red-crowned Ant Tanager (*Habia rubica*: Cardinalidae), a multilocus phylogenetic analysis with emphasis in Mesoamerica

Sandra M. Ramírez-Barrera[1], Blanca E. Hernández-Baños[1], Juan P. Jaramillo-Correa[2] and John Klicka[3]

[1] Departamento de Biología Evolutiva, Facultad de Ciencias, Museo de Zoología, Universidad Nacional Autónoma de México, Ciudad de México, Mexico
[2] Departamento de Ecología Evolutiva, Instituto de Ecología, Universidad Nacional Autónoma de México, Ciudad de México, Mexico
[3] Department of Biology and Burke Museum of Natural History and Culture, University of Washington, Seatle, WA, United States of America

Corresponding author
Blanca E. Hernández-Baños, behb@ciencias.unam.mx

## ABSTRACT

Many neotropical species have a complex history of diversification as a result of the influence of geographical, ecological, climatic, and geological factors that determine the distribution of populations within a lineage. Phylogeography identifies such populations, determines their geographic distributions, and quantifies the degree of genetic divergence. In this work we explored the genetic structure of *Habia rubica* populations, a polytypic taxon with 17 subspecies described, in order to obtain hypotheses about their evolutionary history and processes of diversification. We undertook multilocus analyses using sequences of five molecular markers (ND2, ACOI-I9, MUSK, FGB-I5 and ODC), and sampling from across the species' distribution range, an area encompassing from Central Mexico throughout much of South America. With these data, we obtained a robust phylogenetic hypothesis, a species delimitation analysis, and estimates of divergence times for these lineages. The phylogenetic hypothesis of concatenated molecular markers shows that *H. rubica* can be divided in three main clades: the first includes Mexican Pacific coast populations, the second is formed by population from east of Mexico to Panama and the third comprises the South American populations. Within these clades we recognize seven principal phylogroups whose limits have a clear correspondence with important geographical discontinuities including the Isthmus of Tehuantepec in southern Mexico, the Talamanca Cordillera, and the Isthmus of Panama in North America. In South America, we observed a marked separation of two phylogroups that include the populations that inhabit mesic forests in western and central South America (Amazon Forest) and those inhabiting the seasonal forest from the eastern and northern regions of the South America (Atlantic Forest). These areas are separated by an intervening dry vegetation ''diagonal'' (Chaco, Cerrado and Caatinga). The geographic and genetic structure of these phylogroups describes a history of diversification more active and complex in the northern distribution of this species, producing at least seven well-supported lineages that could be considered species.

## INTRODUCTION

The origins and evolutionary drivers of neotropical diversity are one of the most studied and hotly debated topics since the first biological explorations in the 19th century (*Salvin & Godman, 1879–1904*). Two main hypotheses have been proposed to explain the high levels of neotropical biodiversity (*Rull, 2008*). On one hand, the glacial-interglacial cycles of the Pleistocene (last ~2.6 Mya) have been suggested as the major drivers of divergence, during which allopatric speciation took place in isolated rainforest refuges, particularly during the cooler and drier conditions of the glacial maxima (*Whitmore & Prance, 1987*; *Hooghiemstra & Van der Hammen, 1998*; *Hewitt, 2000*; *Bennett, 2004*). On the other hand, the paleogeographic changes that occurred throughout the Cenozoic (i.e., the last ~66 Mya), like the Andean orogeny and the uprising of the Isthmus of Panama (*Hewitt, 2000*; *Willis & Niklas, 2004*; *Nores, 2004*), have been postulated as major factors driving diversification, as new geographical barriers both increased isolation and promoted divergence. Both hypotheses have however been criticized. For instance, there is little palynological (*Colinvaux et al., 2001*; *Bush & De Oliveira, 2006*), phylogenetic (e.g., *Moritz et al., 2000*; *Glor, Vitt & Larson, 2001*), or ecological-modeling support (*Cowling, Maslin & Sykes, 2001*; *Mayle et al., 2004*) for a Pleistocene origin of tropical diversity. Indeed, most divergence times estimated so far date back to the Neogene, supporting a role for more ancient diversification events. However, differentiation patterns across taxa are highly discordant and frequently cannot be linked to specific landscape or vicariant events (*Burns & Naoki, 2004*; *Burns & Racicot, 2009*; *Mauck & Burns, 2009*) as this hypothesis proposes; instead, they seem more related to each species ability to persist through environmental changes and independently disperse across putative geographic barriers (e.g., *Smith et al., 2014*).

The role of ecological divergence in producing neotropical diversity has taken force in the last decade. For instance, alternating cycles of contracting and expanding tropical forests could have resulted in multiple bouts of differentiation that produced assemblages of "old" and "young" taxa exclusive to each habitat (e.g., *Pennington et al., 2004*; *Smith et al., 2013*). This is not only important for species diverging *in situ*, but also for taxa that dispersed and differentiated across newly available habitats, such as those involved in inter-continental exchange after the uplifting of the Isthmus of Panama. It has been suggested that animals tolerant to a variety of habitats and elevation zones quickly expanded through this corridor into North America, while species restricted to more humid conditions could only disperse more recently into northern latitudes (*Zamudio-Beltrán & Hernández-Baños, 2015*).

Another point that fuels this debate is the species concept itself. That is, evidence supporting one or another hypothesis can be dismissed or not depending on whether the taxa studied are considered (or not) independent species. Traditionally, species boundaries have been drawn based on morphology, but the onset of high-throughput molecular techniques has allowed inferring these boundaries from statistical models based on the
coalescent framework (e.g., Bayesian Phylogenetics and Phylogeography; BP&P; *Yang & Rannala, 2010*). As both views have their own limitations (see for instance *Sukumaran & Knowles, 2017*), an integrative approach based on different types of data (morphological, ethological, ecological, molecular, etc.) seems necessary to operationally determine whether lineages have been evolving separately and therefore can be considered different species (*De Queiroz, 2007*).

The Red-crowned Ant Tanager (*Habia rubica*, Cardinalidae) is a highly polytypic taxon with marked geographical variation; it comprises up to 17 subspecies, (https://avibase.bsc-eoc.org/avibase.jsp?lang=EN) most of which were described based on the variation in the hue and intensity of plumage coloration (*Hilty, 2011*). Its current distribution ranges from central Mexico to north-eastern Argentina and southeastern Brazil, and encompasses regions with very different ecological conditions and/or separated by recognized biogeographical barriers (*Hilty, 2011*). A previous phylogeographic survey (*Lavinia et al., 2015*) suggested that this species originated in South America, where it bears at least two clearly differentiated clades, one in the Atlantic Forest of Brazil and another in the rainforests of the Amazon basin. However, a limited sampling precluded the authors to explore the colonization and diversification within Central America and Southern Mexico.

The objective of this paper is to evaluate the nature and geographic structuring of genetic variation within and among populations of *H. rubica,* using both nuclear and mitochondrial genetic markers in order to provide a phylogenetic hypothesis about the evolutionary history of the species. We used Bayesian and coalescent methods to generate a phylogeny and estimate divergence times between clades, to assess the validity of the genetic groupings identified, using species delimitation methods (*Rannala & Yang, 2003*; *Yang & Rannala, 2010*). Emphasis was made on the missing parts of the biogeographical history of the species.

## MATERIAL & METHODS

### Taxon sampling and laboratory procedures

We sequenced 125 individuals of *H. rubica* species that covered the species distribution, including samples from Mexico (using field collection on permit from Instituto Nacional de Ecología, SEMARNAT: FAUT-0169), Guatemala, El Salvador, Honduras, Nicaragua, Costa Rica, Panama, Peru, Bolivia, Paraguay, Brazil, Argentina and Venezuela (Fig. 1A). We also included 16 samples from the genus *Chlorothraupis* (*C. olivacea*, *C. carmioli* and *C. stolzmanni*), the sister group of *H. rubica* (*Klicka, Burns & Spellman, 2007*), and three congeneric samples that were used as outgroups (*H. fuscicauda*, *H. atrimaxillaris*, and *H. gutturalis*; see Table S1).

We isolated genomic DNA from tissues using Qiagen DNAeasy kit (Qiagen, Valencia, CA, USA) following the manufacturer's protocol. We amplified five molecular markers via the polymerase chain reaction (PCR) in 12.5 µl volumes. The mitochondrial marker *ND*2 (1,041 bp), was generated for all individuals ($n = 144$) using the primers L5219 and H6313 (*Sorenson et al., 1999*). We then amplified four nuclear markers for a subset of individuals with the highest DNA quality ($n = 37$). These nuclear loci included two Z-linked

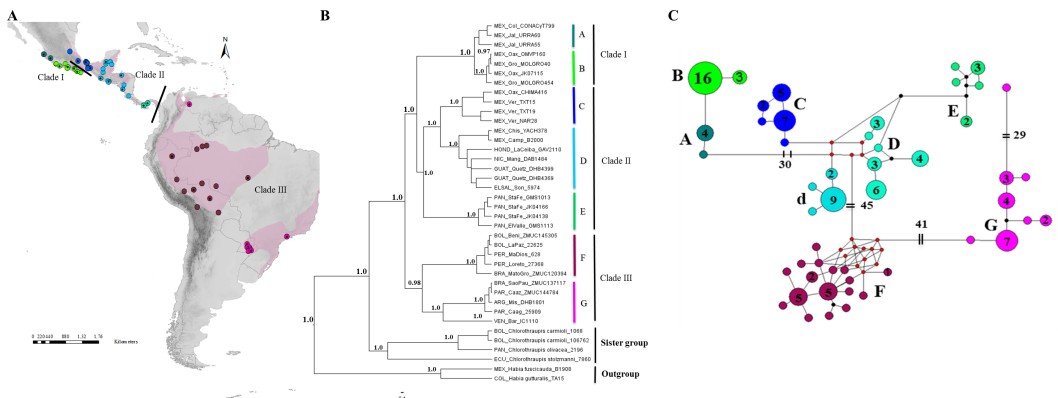

**Figure 1** **Geographical distribution, phylogenetic consensus tree and haplotypes network.** (A) Geographical distribution (indicated by pink shading) and sampling points of the *H. rubica* species; the mitochondrial DNA sampling is represented by dots color and the nuclear DNA sampling is highlighted with a black dot on the dots color. ArcGIS (ArcMAP 10.2.2; Esri, Redlands, CA, USA). (B) Phylogenetic consensus tree representing the relationship among populations from *H. rubica*, based on Bayesian inference from a multilocus dataset. Values above branches denote posterior probabilities (PP). (C) Haplotypes network, where the phylogroup "D" corresponds with individuals from Chiapas-Yucatan peninsula to Costa Rica and "d" corresponds with individuals from Guatemala and El Salvador (the number inside of circles indicate the number of individuals that shared each haplotype).

introns, the aconitase I gene (*ACOI*-I9, 852 bp) amplified using the primers ACOI-I9F and ACOI-I9R (*Kimball et al., 2009*), and the receptor of the tyrosine kinase MUSK gene (*MUSK*, 576 bp) amplified using the primers MUSK-I3F and MUSK-I3R (*Kimball et al., 2009*). We also amplified two autosomal loci, a beta-fibrinogen gene (*FGB*-I5, 577), using the primers Fib5 and Fib6 (*Kimball et al., 2009*) plus the Ornithine Decarboxylase ODC gene (*ODC*, 711 bp) using the primers ODCF and ODCR (*Primmer et al., 2002*). We used the following annealing temperatures: 54 °C (ND2), 56 °C (MUSK), 60 °C (ACOI and FGB-I5) and 65 °C (ODC). PCR success was verified on 1% agarose gels.

DNA sequencing was performed in the High Throughput Genomic Unit Service of the University of Washington. We edited and aligned chromatograms with Sequencher v4.8 (GeneCodes Corporation, Ann Arbor, MI, USA). We corroborated the origin of all of our sequences by combining at least two of the following methods: amplifying overlapping gene segments, amplifying or sequencing one region with different primer sets, sequencing both DNA strands for all amplified fragments, or using multiple individuals of a single species. No evidence of NUMT contamination was found (*Zhang & Hewitt, 1996*; *Sorenson & Quinn, 1998*; *Bensasson et al., 2001*), and all nuclear sequences could be easily aligned with the corresponding genes from other avian species. The overall length of the concatenated mitochondrial and nuclear sequences was ~3,800 bp. All sequences were deposited in FigShare (https://figshare.com/s/fe9f9f6fed1686782f62).

## Phylogenetic analyses

Phylogenies were constructed from two different matrices; the first matrix included 144 sequences of the mitochondrial marker ND2 (hereafter referred to as the mtDNA

dataset), and the second matrix included 37 concatenated sequences of the mitochondrial marker ND2 plus the four nuclear markers describe above (hereafter referred to as multilocus dataset). Phylogenetic analyses were performed for each matrix using a Bayesian Inference method (BI), with independent partitions and evolutionary rates defined for each gene region. The model parameters of nucleotide evolution for each DNA region were obtained with jModeltest v3.8 (*Posada, 2008*) using the Akaike Information Criterion (*Akaike, 1973*). Analyses were conducted in MrBayes v3.2 (*Ronquist et al., 2012*), and consisted of four MCMC chains of 50 million generations each, starting from random trees and using a uniform prior distribution of parameters. Trees were sampled every 250 generations, using Tracer v1.6.0 (*Rambaut & Drummond, 2013*), after discarding the initial 20% as burn-in. The remaining trees were used to construct a majority rule consensus tree with a posterior probability distribution. The final tree was visualized in FigTree v1.2.3 (http//tree.bio.ed.ac.uk/software/figtre/). Finally, we used MEGA v7 (*Kumar, Stecher & Tamura, 2016*) to estimate the pairwise genetic distances between lineages from mitochondrial and nuclear markers, applying a bootstrap of 1,000 iterations; this allowed us to indirectly determine the relative amounts of information provided by the mitochondrial and nuclear markers.

## Diversity and genetic structure

A haplotype network was obtained for the mtDNA dataset with the Median-Joining algorithm available in Network v4.6 (*Bandelt, Foster & Röhl, 1999*). We used DnaSP v5.0 (*Librado & Rozas, 2009*) for estimating the number of haplotypes ($H$) and segregating sites ($S$), the haplotype ($Hd$) and nucleotide ($\pi$) diversities, and the Tajima's $D$ (*Tajima, 1989*) and Fu's $F$ (*Fu, 1997*) values for the genetic clusters recovered in the BI mitochondrial phylogeny mentioned above. Significance for the last two summary statistics was inferred from 1,000 replicates of the coalescent algorithm.

## Species delimitation

The phylogeny obtained from the multilocus dataset was used to assess species delimitations under the Bayesian framework implemented in BP&P (*Yang & Rannala, 2010*). This method allows the estimation of posterior distributions of competing models with differing number of species. It incorporates a reversible-jump Markov chain Monte Carlo (rjMCMC) algorithm and a user-specified tree, where it assigns speciation probability values (*Rannala & Yang, 2003*; *Yang & Rannala, 2010*). BP&P assumes constant population sizes, no gene flow among lineages, and no population structure within lineages after speciation occurred. It furthers includes divergence time ($\tau$) and mutation-scaled effective population size ($\theta$) parameters. We first performed several preliminary runs with the two available algorithms (0: fixed tree, and 1: tree as guide tree) and different seed numbers to confirm consistency and fine-tuning the priors ($\varepsilon$ and $\tau$). After selecting algorithm 0, we conducted a first set of analyses allowing the program to make automatic adjustments to the priors (finetune = 1). Because different values of $\tau$ and $\theta$ can affect the model posterior probabilities for a same guide tree (*Leaché & Fujita, 2010*; *McKay et al., 2013*), and to explore a wide range of speciation histories, we performed a series of simulations using three different parameter

combinations: 1) large population size/deep divergence [G (1, 10) for both $\theta$ and $\tau$], 2) small population size/ shallow divergence [G (2, 2000) for both $\theta$ and $\tau$], and 3) large population size/shallow divergence [G (1, 10) for $\theta$ and G (2, 2000) for $\tau$] (*Leaché & Fujita, 2010*; *Smith et al., 2013*). We then performed a final set of analyses that incorporated the finetune parameters determined in the previous analysis (finetune = 0) with the same three different combinations for $\theta$ and $\tau$ parameters described above. Each set was composed of 100,000 generations, and samples were taken every 5 steps after discarding the initial 50,000 iterations as burn-in. All analyses were performed on the BPPX graphics user interface (http://abacus.gene.ucl.ac.uk/software) using the command line.

### Tests of divergence times

Divergence times were inferred from the multilocus dataset. Analyses were performed in Beast v1.7 (*Drummond et al., 2013*) using the previously defined gene partitions with the corresponding evolutionary rates and models as estimated above (*Ellegren, 2007*; *Smith & Klicka, 2010*) and the seven phylogroups identified with the multilocus dataset. Clade divergence was simulated as a Yule's first tree (*Yule, 1924*) with a lognormal relaxed molecular clock (*Drummond et al., 2006*). Analyses included 100 million iterations, with samples taken every 1,000 generations after eliminating 25% of the trees as burn-in. Mixing and likelihood stability were confirmed with TRACER v1.6 (*Rambaut & Drummond, 2013*) to make sure that the appropriate effective sample size (ESS) was over 200 units for each parameter (*Drummond & Rambaut, 2007*). The best supported tree was chosen with TreeAnnotator 1.8.0 (*Drummond & Rambaut, 2007*) through Maximum Clade Credibility.

Given the absence of fossils for *H. rubica*, or any close relatives, two calibrations based on geological data were used. First, the closing of the Isthmus of Panama, estimated between 3.1 and 4.0 Ma (*Daza, Castoe & Parkinson, 2010*; *Smith & Klicka, 2010*), was used for the separation of South and Central American populations; and second, the uprising of the Talamanca Cordillera near the Costa Rica–Panama border, estimated to have occurred between 2.5 to 3.9 Ma (*Marshall & Liebherr, 2000*; *Daza, Castoe & Parkinson, 2010*), was employed as the time of divergence between populations in Panama and those occurring elsewhere throughout Mesoamerica. In addition, as estimating divergence times based on external event that could be unrelated to a species biology can result in wrong inferences and circular reasoning, we estimated divergence dates a second time but using only the mtDNA dataset (which has a higher sample size) and no calibration points. This could be seen as an independent test of confidence for the estimates obtained for the multilocus analysis with geological constraints. We assumed a mutation rate of $1.25 \times 10^{-2}$ substitutions/site/lineage/Myr (*Smith & Klicka, 2010*), other parameters and priors were kept identical.

## RESULTS

### Phylogenetic analyses

We obtained mitochondrial (1,041 bp) and concatenated (3,757 bp) datasets. The best-fit models for each molecular markers were follows: ND2, TrN+I+G; ACOI-I9, TIM2+G; MUSK, TrN+G; FGB-I5, HKY+G and ODC, TrN.

The phylogenies obtained with mitochondrial and multilocus dataset revealed well-supported topologies, independently of the method employed. Both topologies are composed of three main clades with seven and eight subclades (considered here as "phylogroups") for the multilocus and mitochondrial datasets, respectively. The first clade in the multilocus phylogeny (I) is comprised of two phylogroups with an apparent separation between populations occupying the western (A; Fig. 1B) and southern (B; Fig. 1B) Pacific coast of Mexico. The second clade (II) was composed of three well-supported phylogroups distributed from eastern Mexico to Panama. The first includes all individuals from the Mexican Gulf Coast, west of the Isthmus of Tehuantepec (C; Fig. 1B); the second one comprised samples collected east of this Isthmus and southward into Nicaragua (D; Fig. 1B); and the third one was composed by samples from Panama (E; Fig. 1B). The last clade (III) was composed of (at least) two phylogroups, one with individuals from the Amazon basin (F; Fig. 1B) and another one with birds from eastern South America (G; Fig. 1B). A single sample from Venezuela appears distinctive and is sister to this eastern clade. However, additional samples from Venezuela will be required for a proper systematic assessment.

The mitochondrial phylogeny, although based on many more samples, was almost identical to the one obtained with the multilocus dataset (Fig S1); it did however, show that phylogroup D was composed of two subgroups: one comprised of individuals from the Yucatan Peninsula, Honduras, Nicaragua and Costa Rica, and the other one of individuals from Guatemala and El Salvador.

The pairwise genetic distances showed significant genetic differentiation in the mitochondrial but not in the nuclear markers (Table 1). That is, the genetic distances were smaller for nuclear than for mitochondrial markers (Table 1). This implies that the mitochondrial marker provides larger amounts of relative information than the nuclear gene-regions (Table 2).

## Diversity and genetic structure

The haplotype mtDNA network (Fig. 1C) revealed the same eight groups obtained with BI phylogeny (Fig S1). A high number of haplotypes had to be reconstructed to link some of these groups. For instance, 30 mutational steps separated the two groups from the Mexican Pacific coast from the other clusters, 45 more steps were needed to link the clades from South and Central America, while the two groups from South America were 41 steps apart; an additional 29 mutations were necessary to join the lone sample from Venezuela to the southeastern South America clade.

Overall, diversity estimates differed greatly between mtDNA phylogroups, with the South American and Panamanian ones being the more diverse, and the groups from the Mexican Pacific coast the least genetically rich. Tajimas $D$ and Fu's $F$ values were negative in all phylogroups, excepting phylogroup D (eastern Mexico to northern Central America). However, these values were significant only for the two South American Clades (F & G; Fig. S1); thus suggesting recent population expansions (Table S2).

**Table 1** Corrected pairwise genetic *p*-distances between lineages of *H. rubica* based on mitochondrial and nuclear markers.

| Phylogroups | A | B | C | D | E | F | G |
|---|---|---|---|---|---|---|---|
| | | | | mtDNA | | | |
| A | | | | | | | |
| B | 0.7** | | | | | | |
| C | 4.1* | 4.4* | | | | | |
| D | 4.1* | 4.5* | 1.7** | | | | |
| E | 5.1* | 5.5* | 2.7** | 2.4** | | | |
| F | 6.4* | 6.4* | 5.7* | 5.8* | 6.6* | | |
| G | 6.8* | 6.6* | 5.7* | 5.8* | 6.7* | 5.4* | |
| | | | | nDNA | | | |
| A | | | | | | | |
| B | 0.0** | | | | | | |
| C | 0.1** | 0.1** | | | | | |
| D | 0.1** | 0.1** | 0.1** | | | | |
| E | 0.1** | 0.1** | 0.1** | 0.1** | | | |
| F | 0.2** | 0.2** | 0.1** | 0.2** | 0.2** | | |
| G | 0.6** | 0.6** | 0.5** | 0.6** | 0.6** | 0.4** | |

Notes.
*<0.05.
**<0.005.

**Table 2** Relative amounts of information in the mitochondrial and nuclear DNA.

| Molecular marker | Samples | Sequences | Bases pair | Monomorphic sites | Polymorphic sites | More than two variants | Gaps | Missing data |
|---|---|---|---|---|---|---|---|---|
| Mitochondrial DNA | 125 | 125 | 1,041 | 782 | 259 | 229 | – | – |
| ACOI-I9 | 38 | 76 | 852 | 452 | 112 | 8 | 7 | 281 |
| MUSK | 39 | 78 | 576 | 181 | 54 | 2 | 4 | 337 |
| FGB-I5 | 38 | 76 | 577 | 236 | 60 | 4 | 2 | 279 |
| ODC | 39 | 78 | 711 | 584 | 46 | 1 | 8 | 73 |
| Nuclear DNA | 37 | 74 | 2,716 | 1,733 | 268 | 17 | 20 | 695 |

## Species delimitation

Considering a hypothesis of seven species for this taxon, the Bayesian species delimitation supported the guide tree displaying speciation probabilities of 1.0 in most nodes, independently of the finetune parameter used (finetune = 1 or 0, Figs. 2A and 2B respectively). Seven species were supported with high speciation probabilities (0.97 to1.0) on the tree guide. These species corresponded to the lineages A, B, C, D, E, F and G. The different prior distributions for $\theta$ and $\tau$ had no effect on most nodes, the only exception being that of lineages A and B which were influenced by at least one of the prior combinations (from 0.37 to 0.97). However, while these and other analyses with similar topologies and highly supported clades suggest a multi-species complex within *H. rubica*, other types of data are still necessary to support this hypothesis (*Sukumaran & Knowles, 2017*; see below).

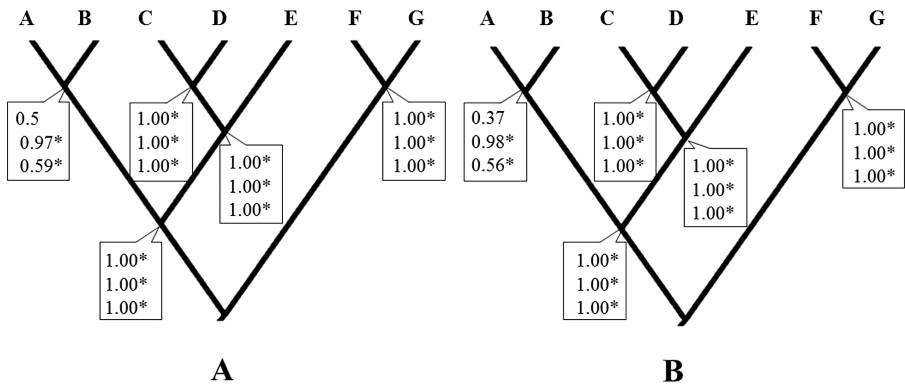

**Figure 2 Bayesian species delimitation.** (A) Bayesian species delimitation results assuming seven species (lineages) from multilocus phylogeny of *H. rubica* (Finetune = 1). (B) Bayesian species delimitation results assuming seven species (lineages) from multilocus phylogeny of *H. rubica* (Finetune = 0, parameters estimated previously). The speciation probabilities are provided for each node under each combination of priors for $\tau$ and $\theta$: top, priors $\theta \sim G(1, 10)$ and $\tau 0 \sim G(1, 10)$; middle, priors $\theta \sim G(2, 2000)$ and $\tau 0 \sim G(2, 2000)$; bottom, priors $\theta \sim G(1,10)$ and $\tau 0 \sim G(2, 2000)$. We consider speciation probability values >0.95 as strong support for a speciation event.

## Tests of divergence times

According the multilocus dataset, the divergence between *H. rubica* and *Chlorothraupis* dated back to the late Neogene, between 6.65 and 3.91 Myr ago (Fig. 3A), and the ensuing diversification of *H. rubica* was rather rapid. The first split was estimated some 3.75 to 3.35 Myr, between the South American (clade III) and Central/Mesoamerican lineages (clades I and II); these two groups split again around 3 Myr ago. In Central America, the first divergence event (3.4–2.7 Myr) occurred between the lineages from the Mexican Pacific coast (Clade I) and the other phylogroups (Clade II). Within clade II, there was a gradual and northward divergence of lineages from Panama to eastern Mexico; these three diversification events during the last ~3Myr or ~2Myr. An independent analysis performed on the mtDNA dataset and with no calibration constraints, produced an identical topology with older mean divergence times (i.e., divergence within *H. rubica* occurred during the last 5.6 Myr). However, as confidence intervals were much wider (Fig. 3B) and overlapped with those obtained with the multilocus dataset, they provide support for the dates estimated based on geological times.

## DISCUSSION

Our results show that *H. rubica* has considerable geographic genetic structure. We found seven lineages through its distribution; five of them are in Mesoamerica, where the evolutionary history has been more dynamic and complex, and two in South America (this last results is similar with was found by *Lavinia et al., 2015*). Most lineages are delimited by recognized geographic and ecological barriers, like the Rio Balsas depression in Mexico, the Central American Volcanic Arc, the Isthmus of Panama in Central America, or the dry vegetation diagonal in South America. We recovered a reciprocally monophyletic

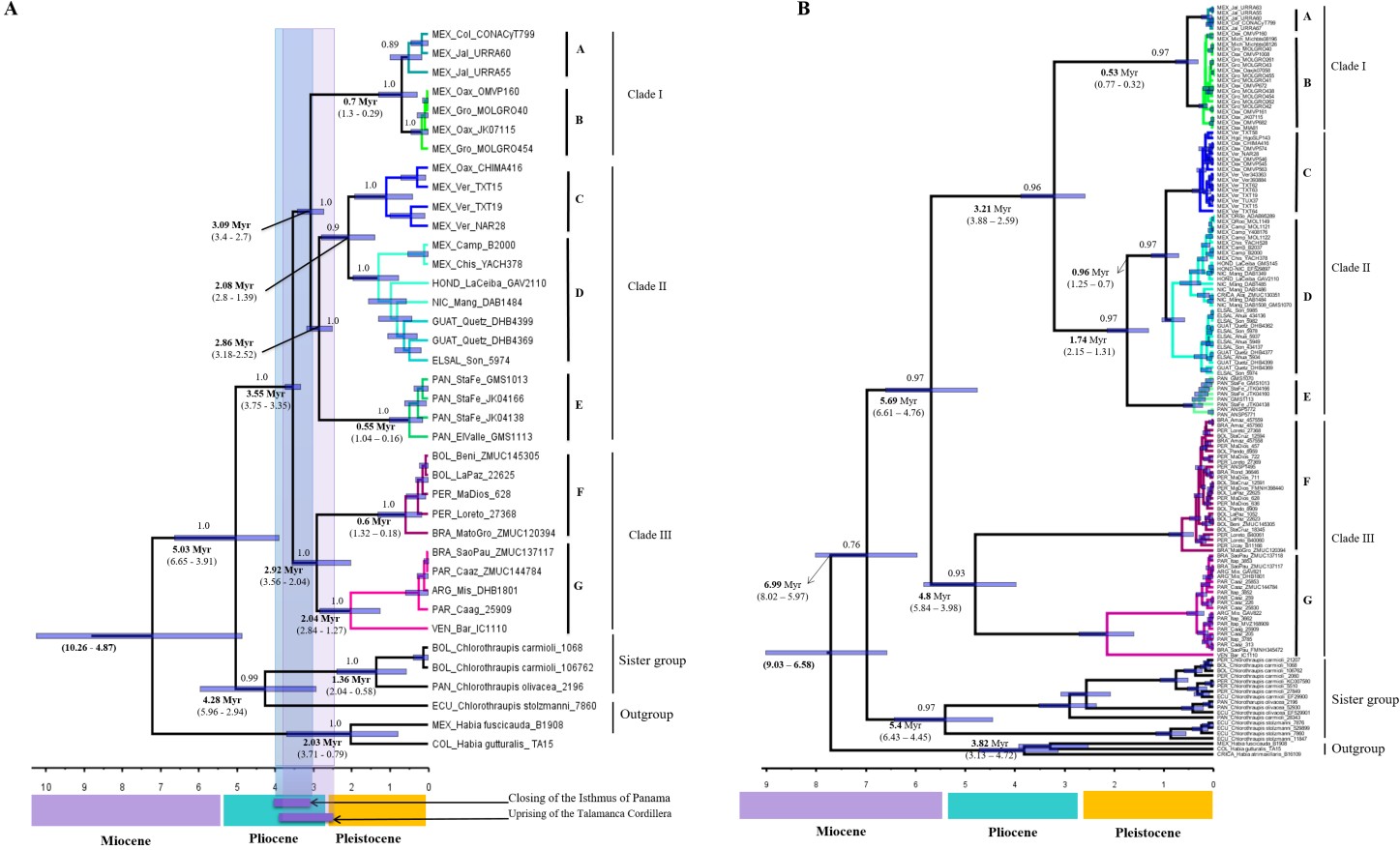

**Figure 3** **Divergence times trees.** (A) For multilocus dataset (confidence interval values are inside parentheses), calibration points are shown in the time bar. (B) For ND2 dataset, we use sustitution rates of $2.5 \times 10^{-2}$ substitutions/site/lineage/Myr.

relationship between *H. rubica* and the genus *Chlorothraupis*, which coincides with previous studies (*Klicka, Burns & Spellman, 2007*; *Klicka, Johnson & Lanyon, 2000*; *Barker et al., 2015*; *Lavinia et al., 2015*); therefore the genus *Habia* must be considered paraphyletic.

## Phylogeny, diversity and genetic structure

The mitochondrial DNA topology indicates that populations of Mexico, Central and South America are profoundly differentiated. In this topology eight phylogroups were identified, six in Mesomerica and two in South America (Fig S1). The relationship between mitochondrial haplotypes could be determined by the action of various historical processes that promoted a deep genetic structure in Mesoamerica, resulting in a clear example of cryptic speciation. This complicated phylogenetic structure in Mesoamerica is consistent with the complex geological and biogeographic history of the region (*Coates & Obando, 1996*), and the emergence of geographic features such as the Rio Balsas depression, the Isthmus of Tehuantepec in southern Mexico, the Motagua-Polochic-Jocotán fault system in the North of Central America, and the Talamanca Cordillera in Costa Rica, which presumably played roles in shaping the observed geographic and genetic patterns

(*Daza, Castoe & Parkinson, 2010*; *Gutiérrez-García & Vázquez-Domínguez, 2012*; *Gutiérrez-García & Vázquez-Domínguez, 2013*; *Suárez-Atilano, Burbrink & Vázquez-Domínguez, 2014*). Our results together with the study of *Lavinia et al. (2015)* provide a robust hypothesis of the evolutionary history of *H. rubica*. For instance, our study confirms the main results of *Lavinia et al. (2015)* for South America, and improves the resolution for Mesoamerica given our larger sample size and wider area covered.

In South America, our results, in combination with those of *Lavinia et al. (2015)*, showed two distinct phylogroups that reflected population divergence according to the habitats preferred by *H. rubica* (*Pennington et al., 2004*; *Pennington, Lavin & Oliveira-Filho, 2009*; *Lavin, 2006*; *Smith et al., 2013*; *Smith et al., 2014*; *Pennington & Lavin, 2016*): one western phylogroup distributed in the mesic forest of the Amazon basin, and another one in the seasonal forests of the eastern and northern parts of South America. This last lineage included the single sample from Venezuela, instead of the western clade that is geographically more proximate (*Prates et al., 2016*; *Prates et al., 2017*). This could be indicating that the deep evolutionary history of the seasonal forests in South America (*Pennington et al., 2004*; *Pennington, Lavin & Oliveira-Filho, 2009*) are driving the divergence of other taxa. Additional sampling in this region is thus necessary to test this hypothesis.

The multilocus DNA topology identified seven of the eight phylogroups found with the mitochondrial DNA (Fig. 1B), in all cases the phylogroups are the same with the exception of an additional separation of a Yucatan Peninsula and northern of Central America clade identified using the more rapidly evolving mtDNA marker. All identified phylogroups (mitochondrial and multilocus dataset) are consistent with clear geographical discontinuities and are supported with high posterior probabilities.

Although, the haplotype network recovered the same groups identified in the mitochondrial DNA phylogeny (Fig. 1C), it was necessary to invoke five hypothetical haplotypes between Mexican and Central American groups. These results support the idea about the complex geological history in Mesoamerica resulting in a high degree of genetic structure in the northernmost distribution of this species. In South America, 12 hypothetical haplotypes were required, but unlike the previous case, this could be explained by the lack of sampling in this specific area.

The genetic distances between phylogroups were relatively high and supported the groups identified in our phylogenetic and BPP analyses. The largest difference found was between the group from the Western Pacific Coast of Mexico (group A) and the group from the Southeast of South America (group G) (Table 1). This result is not surprising because the geographical distance between them is also the largest and is congruent with the divergence date (Figs. 3A and 3B). On the other hand, the more similar phylogroups were the Western (A) and the Southern Pacific Coast of Mexco (B) groups, which represent the most recent divergence event (Figs. 3A and 3B). As expected the mitochondrial DNA provides more polymorphic sites because its rate of evolution is faster (Table 2) (*Brown, George Jr & Wilson, 1979*).

## Species delimitation and divergence times

The deep divergence times among lineages estimated herein (Figs. 3A and 3B), most of which dated back to Pleistocene and Pliocene, support the criticisms raised regarding the allopatric speciation hypothesis that invoked the Pleistocene glacial cycles as the major drivers of neotropical bird diversity and closing of the Isthmus of Panama (e.g., *Moritz et al., 2000*; *Cowling, Maslin & Sykes, 2001*; *Glor, Vitt & Larson, 2001*; *Mayle et al., 2004*). According to the ND2 divergence analyses, the first split was during the Miocene (5.69 Myr), and this is the main discordance with respect to the calibrated multilocus divergence analyses (3.55 Myr). These dates are also consistent with the compelling evidence for speciation drawn from the BP&P analyses, which suggest that enough differences have accumulated between most of the *H. rubica* lineages to consider them a complex of cryptic species (Fig. 2).

At present, intraspecific delimitation within *H. rubica* is based on geographic variation in plumage color, with 17 subspecies currently recognized (https://avibase.bsc-eoc.org/avibase.jsp?lang=EN). Many of these subspecies are not supported by the lineages described herein (Fig. 2). For instance, the subspecies *rosea* and *rubicoides* are paraphyletic (lineages A&B, and C&D, from BP&P analysis, respectively). The first taxon is further intermingled with ssp. *affinis* individuals in lineage B, while the second one is combined with individuals of ssp. *holobrunnea* and *nelsoni* in the lineages C and D. Some subspecies also appear indistinguishable with the markers that we used (e.g., ssp. *hesterna*, *peruviana* and *rhodinolaema* within lineage F). Overall, our results support a taxonomic revision of *H. rubica*.

Our test for species delimitation suggests that *H. rubica* should minimally be split into the following species: (1) *H. rosea*, distributed along the Pacific coast of western Mexico (Jalisco, Nayarit and Colima; lineage A); (2) *H. affinis*, distributed along the Pacific coast of southern Mexico (Michoacan, Guerrero and Oaxaca; lineage B); (3) *H. holobrunnea*, distributed over the slope of Gulf of Mexico (lineage C); (4) *H. rubicoides*, distributed from eastern Mexico to southern Nicaragua (lineage D); (5) *H. vinacea*, allopatric in Panama (lineage E); (6) *H. rhodinolaema*, distributed in the Amazon basin (lineage F); and (7) *H. rubica*, distributed in eastern Brazil, Argentina, and Paraguay (lineage G). Further tests and sampling should be necessary for a more adequate description of the Venezuela's individuals, currently classified as spp. *perijana* (*Hilty, 2011*).

There are criticisms to the BP&P analysis implemented herein. For instance, simulation data have shown that it tends to misidentify population structure as species boundaries (*Sukumaran & Knowles, 2017*). However, it must be noted that our delimitation not only detected genetics lineages, but lineages whose distribution coincides with the geographical barriers and ecological differences (habitat preference) of each taxon. That is, it highlights the role of other factors, like vicariance, dispersal and ecology, under a framework of ecological allopatric speciation, which is one the drivers proposed for generating bird species diversity in the neotropics (*Hooghiemstra & Van der Hammen, 1998*; *Hewitt, 2000*; *Bennett, 2004*; *Smith et al., 2013*). It also shows that this bird diversity is underestimated, and that species cannot be described or estimated based only on morphological criteria. From this point of view, diversifying the techniques for

species delimitation, like incorporating bioacoustics, high-throughput sequencing, niche modelling and advanced statistical tools, will also help addressing consequential questions in taxonomy and biogeography. For instance, such multifactorial data should help determining operationally whether the *H. rubica* lineages inferred herein are indeed evolving separately (see *De Queiroz, 2007*).

## CONCLUSIONS

The phyogenies obtained independently with mitochondrial and multilocus datasets revealed well-supported topologies for the polytypic taxon *H. rubica*. Both topologies are composed of three main clades with eight and seven phylogroups (lineages), respectively. The mtDNA haplotype network produced the same eight groups obtained with the mitochondrial data. The Bayesian species delimitation analysis found seven species highly supported: (1) *H. rosea*, distributed along the Pacific coast of western Mexico; (2) *H. affinis*, distributed along the Pacific coast of southern Mexico; (3) *H. holobrunnea*, distributed distributed over the slope of Gulf of Mexico; (4) *H. rubicoides*, distributed from eastern Mexico to southern Nicaragua; (5) *H. vinacea*, allopatric in Panama; (6) *H. rhodinolaema*, distributed in the Amazon basin and (7) *H. rubica*, distributed in eastern Brazil, Argentina and Paraguay. Further tests and sampling should be necessary for a more adequate description of the samples from Venezuela, currently classified as spp. *perijana*. The species subdivision of *H. rubica* coincides with geographical barriers and ecological differences between taxa, highlighting the role of vicariance, dispersal and ecology in bird species diversification in the neotropics.

## ACKNOWLEDGEMENTS

We thank the following institutions for providing samples: Museo de Zoología ''Alfonso L. Herrera (UNAM)'', The Burke Museum (UWBM), El Colegio de la Frontera Sur (ECOSUR-Ch), the Louisiana Museum of Natural History (Louisiana State University) and the Natural History Museum (University of Kansas). We are also grateful to Isabel Vargas Fernández, Raul Ivan Martínez, Luz Estela Zamudio Beltrán, Ernesto Espinosa Jaramillo, Alejandro Gordillo Martínez and Fabiola Ramírez Corona for technical help.

### Funding

The research was supported by PAPIIT/DGAPA, UNAM through a grant to Blanca E. Hernández-Baños (IN215614). Sandra M. Ramírez-Barrera was supported by a scholarship from CONACyT, Mexico. This paper was written during the sabbatical leave of Blanca E. Hernández-Baños at the Museo Nacional de Ciencias Naturales, Madrid, supported by PASPA/DGAPA UNAM and CONACyT (Mexico). Juan P. Jaramillo-Correa was on a sabbatical leave during the writing of this paper at UMR INRA-Université de Bordeaux, supported by PASPA/DGAPA UNAM. The funders had no role in study design, data collection and analysis, decision to publish, or preparation of the manuscript.

## Grant Disclosures

The following grant information was disclosed by the authors:
PAPIIT/DGAPA, UNAM: IN215614.
CONACyT, Mexico.
PASPA/DGAPA UNAM.
UMR INRA-Université de Bordeaux.

## Competing Interests

The authors declare there are no competing interests.

## Author Contributions

- Sandra M Ramírez-Barrera conceived and designed the experiments, performed the experiments, analyzed the data, prepared figures and/or tables, authored or reviewed drafts of the paper, approved the final draft.
- Blanca E Hernández-Baños conceived and designed the experiments, performed the experiments, contributed reagents/materials/analysis tools, authored or reviewed drafts of the paper, approved the final draft.
- Juan P. Jaramillo-Correa and John Klicka authored or reviewed drafts of the paper, approved the final draft.

## Field Study Permissions

The following information was supplied relating to field study approvals (i.e., approving body and any reference numbers):

The field collection permit was provided by Instituto Nacional de Ecologia, SEMARNAT, Mexico (FAUT-0169).

## Data Availability

Ramirez, Marisol (2018): *Habia rubica* sequences. figshare. Dataset. https://doi.org/10.6084/m9.figshare.5926591.v1.

## Supplemental Information

Supplemental information for this article can be found online at http://dx.doi.org/10.7717/peerj.5496#supplemental-information.

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
