# Peer review of "Deep divergence of Red-crowned Ant Tanager (Habia rubica: Cardinalidae), a multilocus phylogenetic analysis with emphasis in Mesoamerica"

_PeerJ, doi:10.7717/peerj.5496_

## Round 0.1 · original submission · Major Revisions

Dear authors

Our reviewers were quite critical of your ms. They ask for a better discussion of a recent paper in MPE on the same topic. If you are willing to rewrite and refocus your ms, we can reconsider it. But be aware that the same critical reviewers will judge your ms.

Kind regards

Michael Wink
Academic editor

Reviewer 1 ·

Basic reporting

The authors present their research on the phylogeographic history of the Neotropical bird species complex, Habia rubica. The sampling covers the entire geographical distribution of the species. However, a very similar study of the same species, which is cited by the authors (Lavinia et al. 2015), also used samples covering the entire distribution. In that study, the species complex was most strongly sampled in the Atlantic Forest and Amazonia, while the present study sample favors Central America. Lavinia also utilized a much more complete data set (DNA sequencing, morphological and vocal analysis) than found in the present study which relies exclusively on DNA sequencing. Therefore, this paper does not contribute a great deal of new information to what is already known about the taxonomy and biogeographical history of Habia rubica across its entire distribution. The most important contribution of this paper is the extensive sampling in Central America, reveling new clades not described by Lavinia. However the discussion is weak, especially since the authors do not develop a comparison with the findings of Lavinia. I feel that this manuscript has the potential to be an important and interesting contribution if the authors can refocus and reexamine their data to a regional scale, restricted to biogeographic issues of the species in Central America that were not sufficiently described by Lavinia. The title should be modified giving emphasis to phylogeography on a regional scale (Central America). I also suggest not emphasizing taxonomy since the only method used was DNA sequencing, unlike the more extensive morphological and vocal analysis found in Lavinia.

Experimental design

Line 153 - How was convergence in the Bayesian run determined? Many authors simply use TRACER, but this may not be sufficient. I suggest using AWTY in addition to TRACER to determine if the tree topology had reached convergence. While overall tree likelihoods and individual model parameters reach convergence in less than 3 million generations, the tree topology requires up to 8 million generations to reach convergence. In general, I suggest using a minimum of 20 or even 50 million generations for Bayesian analyses in MrBayes and then using a very generous burn-in. In contrast, you use a 100 million generation sampling strategy for BEAST which seems good!

Line 197 – Why Yule process? Yule process assumes that there are no extinctions, which could occur here.

Validity of the findings

no comment

Additional comments

The authors present their research on the phylogeographic history of the Neotropical bird species complex, Habia rubica. The sampling covers the entire geographical distribution of the species. However, a very similar study of the same species, which is cited by the authors (Lavinia et al. 2015), also used samples covering the entire distribution. In that study, the species complex was most strongly sampled in the Atlantic Forest and Amazonia, while the present study sample favors Central America. Lavinia also utilized a much more complete data set (DNA sequencing, morphological and vocal analysis) than found in the present study which relies exclusively on DNA sequencing. Therefore, this paper does not contribute a great deal of new information to what is already known about the taxonomy and biogeographical history of Habia rubica across its entire distribution. The most important contribution of this paper is the extensive sampling in Central America, reveling new clades not described by Lavinia. However the discussion is weak, especially since the authors do not develop a comparison with the findings of Lavinia. I feel that this manuscript has the potential to be an important and interesting contribution if the authors can refocus and reexamine their data to a regional scale, restricted to biogeographic issues of the species in Central America that were not sufficiently described by Lavinia. The title should be modified giving emphasis to phylogeography on a regional scale (Central America). I also suggest not emphasizing taxonomy since the only method used was DNA sequencing, unlike the more extensive morphological and vocal analysis found in Lavinia.

A few more comments:
Abstract
Line 32 – I suggest adding more information about Habia rubica, when is mentioned for the first time.

Methods
Line 153 - How was convergence in the Bayesian run determined? Many authors simply use TRACER, but this may not be sufficient. I suggest using AWTY in addition to TRACER to determine if the tree topology had reached convergence. While overall tree likelihoods and individual model parameters reach convergence in less than 3 million generations, the tree topology requires up to 8 million generations to reach convergence. In general, I suggest using a minimum of 20 or even 50 million generations for Bayesian analyses in MrBayes and then using a very generous burn-in. In contrast, you use a 100 million generation sampling strategy for BEAST which seems good!

Line 197 – Why Yule process? Yule process assumes that there are no extinctions, which could occur here.

Results – no comments
Discussion – Should be refocused to a regional scale with a reduced stress on taxonomy.

Reviewer 2 ·

Basic reporting

There are some small editorial issues to address. A good copy editor would help. Comma use is sometimes hard to follow. Fig. 3 tip labels have low resolution.

Experimental design

Sampling is generally good. Only the Venezuela population is a problem, but that is noted in the results.

Validity of the findings

The authors study the genetic structure of populations of a Neotropical bird, Habia rubica. They use one mtDNA and four nuclear loci to produce a tree, to do species delimitation using BPP, and to date divergence times. The results are concordant with major geological features in the taxon’s range, and they find seven lineages that they conclude are species.

The authors used 125 H. rubica from most of the range and used appropriate outgroups. 37 of these were also sequenced at 4 nuclear loci. It is a good study and with a few improvements will better.

One concern is that the authors are using a very simplistic concept of species – what we might term a BPP species. Is this concept widely accepted in ornithology? Barcoding “species” were not considered acceptable to ornithology when I last paid attention. This is not very different from that. In fact, a recent paper that the authors do not cite has done simulation studies to show that BPP will often misdiagnose populations as species (Sukumaran and Knowles, 2017, PNAS 114: 1607).

This leads to the next issue, which is to disregard all other data except the genetic data. It is touched on that the phenotypes disagree with the reconstructed tree, but little is made of this, either in discussing why we might expect 17 phenotypes found to not match 7 genetic lineages or to provide more evidence for the recommended taxonomy. In this the authors go directly against the recommendation of the PNAS paper cited above.

It would be good to know how much genetic variation is present within the in-group for the markers used. We have seen a number of cases in which mtDNA has not provided an accurate phylogeny, and these cases have only come to light with the addition of substantial amounts of nuclear data. It is good that the authors have four nuclear loci (2,716 bp) against the 1,041 bp of mtDNA, but we do not get a summary of how much information is in those loci for the in-group questions, so the mtDNA are still likely driving a lot of the relationship that is reconstructed. In a quick scan of the data on figshare it looks like the nuclear loci are not very information rich for the in-group compared with mtDNA, so quantifying that would help the reader.

A key assumption in dating nodes on the phylogeny is that the species historic distribution was affected by the two geological events used to base divergence estimates. This is a big assumption. It is also not necessary to rely on it alone, because the mtDNA gene ND2 has been extensively studied and has its own mutation rate. This provides an independent check and would be easy to implement.

So I think there are three major issues: 1) the species concept used and the published evidence that BPP oversplits by recognizing populations as species. 2) the relative amounts of information in the mtDNA and nuclear DNA for the ingroup (easily summarised). 3) using ND2 for a second set of divergence estimates so you don’t have to rely alone on the assumption that the species was present and affected by the two geographic events used currently to date the nodes.

In addition, there are some minor issues:

Why figshare? This is an unusual place to archive sequence data. Having the alignments used is useful, but it would also be useful to archive sequences on GenBank with the museum voucher numbers.

I notice that the Lavinia et al. 2015 paper recommended recognition of three species, whereas here the recommendation is seven. That should be discussed more because they included phenotypic analyses. (It also looks like the Venezuelan population would almost certainly emerge with more individuals as another BPP species.)

The Lavinia paper also does a better job with geography in the text and their figure 1. The authors might look at that because I think their treatment could be clearer.

Additional comments

See above.

Reviewer 3 ·

Basic reporting

A Multilocus phylogenetic analysis provides evidence of deep divergence between populations of Red-crowned Ant Tanager (Habia rubica: Cardinalidae).
The authors claim to investigate the genetic structure of Habia rubica populations using a multilocus dataset, five molecular markers were employed and a total of 4.798 nucleotides were used in the analyses. They hypothesized that the species form three main clades: 1-Mexican Pacific coast populations, 2- east of Mexico to Panama, 3-South American populations. The lineages identified within the clades were delimited by geographical barriers in Central and South America and a strong influence of the morphoclimatic domains of South America separating the populations too. The lineages were not congruent with the previously described subspecies. The authors suggested a taxonomic reformulation for Habia rubica.

Experimental design

The methodology was adequated. I suggest few modifications:
1# The information from lines 106 to 110 can be placed in the acknowledgments.
2# I understand that the authors calculated the demographic tests for the different lineages, however, populations from different localities were placed together, which makes the information of table 1 less representative, once the authors mixed different populations together. Therefore, I suggest moving this table 1 to supplemental information.
3# Could you please include the pairwise mismatch p-distances between the lineages? I want to see how distant are the lineages, and if the distance is around 4 to 5%, what is normally observed at species level.

Validity of the findings

Despite the interesting results and the strong support of the phylogeny, the manuscript seeks to investigate the evolutionary history of the Ant Tanager, that have been already published in a recent study “Lavinia et al. 2015 Continental-scale analysis reveals deep diversification within the polytypic Red-crowned Ant Tanager (Habia rubica, Cardinalidae). Molecular Phylogenetics and Evolution, 89:182-193.” Both studies have similar objectives, however, the data analyses were based on different molecular markers and the authors include more samples and more localities, which, I think, reinforce the importance of the results. Overall, both manuscripts displayed of a similar outcome. Then, I think the article is adequate for publication by the PeerJ journal, but it would also be more interesting for publication in an ornithological-specific journal.
Additionally, I was really intrigued by the composition of Clade G: Southwest Brazil, Paraguay, Argentina and Venezuela (Line 231, fig. 1). How the authors hypothesize this close relationship between those populations? Is there a possibility to mistakenly grouping these lineages together? That might also reinforce the status of this lineage as a taxonomic unit with a distinct biogeographic history.

Additional comments

Lines 290-296, The authors could add the time of emergence of the geographical barriers in order to trace a parallel together with the molecular time estimation proposed to the studied species.
Extra space on line 267.

---

## Round 0.2 · Minor Revisions

Dear authors

Please consider the comments of reviewer 2 more seriously. Do not write a long rebuttal; make the changes in the ms.

Kind regards

MWink

Reviewer 1 ·

Basic reporting

No comment

Experimental design

No comment

Validity of the findings

No comment

Additional comments

The authors have now dealt with all my comments on the first round of revision and I feel the manuscript is greatly improved. The revised manuscript clarifies many of the issues in the earlier version, so it is much stronger now. I have no major concerns with this paper

Reviewer 2 ·

Basic reporting

Fail. Some previously addressed problems remain and a new one appears (wrong definition of Pleistocene in its beginning date). Details in General Comments below.

Experimental design

Pass.

Validity of the findings

Conclusions remain problematic in some respects. Details in General Comments below.

Additional comments

PeerJ “Deep divergence of Red-crowned Ant Tanager...”

This is the second time that I have read this manuscript. The authors have made some improvements. There are still some places where it might be improved, including where efforts are given in the response but not added to the paper.

Geographic isolation is usually not a criterion for species delimitation. It is easy for isolated populations to become genetically distinct, but that is not usually enough to recognize them as species. Also, you cite de Quieroz (2007) in the response, but not in the paper. It is important to tell the reader what your framework is and how your results do or do not match it.

I am surprised that the authors chose not to cite the Sukumaran & Knowles 2017 PNAS paper. It shows very clearly one of the major weaknesses of these kinds of BPP analyses and that oversplitting is an expected result. The authors themselves respond that they “are well aware the BPP tends to over estimate the number of true species.” But we do not have a presentation of how the authors determined that their results are not such a case of oversplitting. It is interesting to see a recommendation (paragraph at line 357) that we can’t use “only morphological criteria” for species delimitation, but that the authors view of the future seems to exclude it, which seems opposite to the current direction of taxonomy in most groups (integrative).

My earlier suggestion on independently estimating dates using ND2 as a separate check on the dates obtained using geological events alone (which require key assumptions the birds may not have followed) was not directly included. The response says that the estimates were made, but the Methods do not include information on the mutation rate chosen and the Results do not include mtDNA divergence estimates in the Divergence times section. The response goes on to say that the topology is similar and the clades are well defined, but this is not the point. Dating is. What are the ND2 dates, and how do they compare to the ones that are presented? These mtDNA estimates should be present if only to be able to make direct comparisons to the many previous studies only using mtDNA to study these splitting events across these major geographic barriers. The problem becomes most apparent when we examine Fig. 3, in which the two biogeographic events used for the molecular calibration are highlighted. Because they are the drivers of the relationship of the tree to the timeline, the results are more than a little circular. Use ND2 and give us more confidence. Okay: I see now inclusion of the p-distances in response to Reviewer 3 (Table 1). But please use these to tell us what key node dates are on Fig. 3.

Examination of Fig. 3 also indicates to me that the authors are not using the correct definition of the Pleistocene, which began about 2.6 Mya. If Fig. 3 is correct on its dates, then most of the 7 clades (A-G) probably did arise in the Pleistocene, and all of them may have. The Discussion (lines 332-336) should be adjusted to this.

Some additional issues:

The clades A-G are not aligned well with the tree in Fig. 3.

The table included in the response “Table #” should be added to the paper. So should the Phylogroups table. I would put the former in the main text and move the other one to Supplemental Information.

Reviewer 3 ·

Basic reporting

The authors included the suggestions made by the reviewers in the new version of the manuscript. I believe that this article can be published in the Peer J journal.

Experimental design

The experimental design attended the expectations.

Validity of the findings

The main findings of this study was the improvement of the resolution of the evolutionary history of the populations from Habia rubica distributed along the Mesoamerica area, additionally their dataset brings a larger sample size and wider area covered when compared with the study of Lavinia et al. (2015). Although, the study from Lavinia published in 2015 brings similar results as proposed by the authors in their study.

The well-supported phylogenetic tree and the pairwise p-distance suggest the validity of the results and proposition of taxonomic review for Habia rubica, what can be further validated together with bioacustic and NGS analyses.

Additional comments

No comments.

---

## Round 0.3 · accepted · Accept

Dear authors

Thanks for revising the ms. Good news, your revision is adequate and we can accept your ms.

Thanks for submitting your work to PeerJ.

Greetings
Michael Wink
Academic Editor

#